# Low-Level Expression of p-S6 Is Associated with Nodal Metastasis in Patients with Head and Neck Cutaneous Squamous Cell Carcinoma

**DOI:** 10.3390/ijms25084304

**Published:** 2024-04-13

**Authors:** Celia Gómez-de Castro, Raquel Santos-Juanes, Borja Nuñez-Gómez, Iván Fernández-Vega, Blanca Vivanco, Adela Fernández-Velasco, Sebastián Reyes-García, Jimena Carrero-Martín, Juana M. García-Pedrero, Juan P. Rodrigo, María del Carmen González-Vela, Jorge Santos-Juanes, Cristina Galache

**Affiliations:** 1Grider, Grupo de Investigación en Dermatología, Universidad de Oviedo, 33006 Oviedo, Spain; celiagomez_88@hotmail.com (C.G.-d.C.); vivancoblanca@gmail.com (B.V.); alonsoadela@uniovi.es (A.F.-V.); cristinagalache@gmail.com (C.G.); 2Dermatology Division, Hospital Universitario Central de Asturias, 33011 Oviedo, Spain; sebastian021314@gmail.com (S.R.-G.); jimenacarrerom@gmail.com (J.C.-M.); 3Dermatology Area, Department of Medicine, University of Oviedo, 33006 Oviedo, Spain; raquel.santosjuanes@gmail.com (R.S.-J.); netborja@gmail.com (B.N.-G.); 4Department of Pathology, Hospital Universitario Central de Asturias, Biobank of the Principality of Asturias (BioPA), 33011 Oviedo, Spain; fernandezvivan@uniovi.es; 5Department of Pathology, University of Oviedo, 33006 Oviedo, Spain; 6Instituto de Investigación Sanitaria del Principado de Asturias, 33011 Oviedo, Spain; juanagp.finba@gmail.com (J.M.G.-P.); jprodrigo@uniovi.es (J.P.R.); 7Instituto Universitario de Oncología del Principado de Asturias, University of Oviedo, 33006 Oviedo, Spain; 8Centro de Investigación Biomédica en Red de Cáncer (CIBERONC), Instituto de Salud Carlos III, 28029 Madrid, Spain; 9Department of Otolaryngology-Head and Neck Surgery, Hospital Universitario Central de Asturias, 33011 Oviedo, Spain; 10Department of Pathology, Hospital Universitario Marqués de Valdecilla, 39008 Santander, Spain; carmengonzalezvela@gmail.com

**Keywords:** cutaneous squamous cell carcinoma, p-S6, metastasis

## Abstract

Cutaneous squamous cell carcinoma (cSCC) is the second most common form of skin cancer. The incidence of metastasis for cSCC is estimated to be around 1.2–5%. Ribosomal protein S6 (p-S6) and the p21 protein (p21) are two proteins that play central roles in other cancers. These proteins may be equally important in cSCC, and together, these could constitute a good candidate for metastasis risk assessment of these patients. We investigate the relationship of p-S6 and p21 expression with the impact on the prognosis of head and neck cSCC (cSCCHN). p-S6 and p21 expression was analyzed by immunohistochemistry on paraffin-embedded tissue samples from 116 patients with cSCCHN and associations sought with clinical characteristics. Kaplan–Meier estimators and Cox proportional hazard regression models were also used. The expression of p-S6 was significantly inversely associated with tumor thickness, tumor size, desmoplastic growth, pathological stage, perineural invasion and tumor buds. p21 expression was significantly inversely correlated with >6 mm tumor thickness, desmoplastic growth, and perineural invasion. p-S6-negative expression significantly predicted an increased risk of nodal metastasis (HR = 2.63, 95% CI 1.51–4.54; *p* < 0.001). p21 expression was not found to be a significant risk factor for nodal metastasis. These findings demonstrate that p-S6-negative expression is an independent predictor of nodal metastasis. The immunohistochemical expression of p-S6 might aid in better risk stratification and management of patients with cSCCHN.

## 1. Introduction

Cutaneous squamous cell carcinoma (cSCC) is the second most common form of skin cancer and is newly diagnosed in nearly two million people each year [1,2]. The incidence of metastases for cSCC is estimated to be around 1.2–5% [3]. There are certain characteristics of a primary lesion that imply a higher risk of metastases: primary lesion diameter >2 cm; tumor thickness >6 mm; tumor on or around the ear, lip or temple; recurrent lesions; poorly differentiated grade; desmoplastic growth; tumor budding; microvascular, lymphatic, or perineural invasion; advanced age; and a cSCC in an immunocompromised host [4,5]. cSCC is currently poorly characterized at the molecular level due to the high mutational burden of the disease [6].

Ribosomal protein S6 (p-S6) and p21 (also known as CDKN1A) are two proteins known to play central roles in other cancers. These proteins may be equally important in cSCC, and, together, these could constitute a good candidate for the metastasis risk assessment of these patients.

Progression through the cell cycle in eukaryotic cells is regulated by a suite of cyclin-dependent kinases (CDKs) and CDK inhibitors. p21 is encoded by the CDKN1A gene, and is a member of the CIP/KIP family of CDK inhibitors, together with CDKN1B (p27) and CDKN1C (p57) [7]. The mammalian target of rapamycin complex 1 (mTORC1) promotes various cellular processes, including protein synthesis, proliferation, cell survival, ribosome biogenesis, angiogenesis, migration, invasion, and metastasis by phosphorylation of ribosomal protein S6 kinase (p-S6K; also known as RPS6KB) and eukaryotic initiation factor 4E (eIF4E)-binding protein 1 (4EBP1) [8].

The aim of our study was to analyze p-S6 and p21 in a large series of patients with cSCC of the head and neck (cSCCHN), and to identify associations with clinicopathological features and their impacts on patient prognosis.

## 2. Results

In total, 116 patients of white ethnicity were enrolled in the study (Table 1 and Table 2), 89 of whom were men (76.7%). The mean age of the whole sample was 78.4 years. The mean age and standard deviation (SD) of patients with p-S6 positive tumors was 78.41 ± 8.87 years, and that of the p-S6-negative subgroup was 79.11 ± 7.73 years (Table 1).

The mean age and SD of patients with p21-positive tumors was 78.82 ± 8.27 years, and that of the p21-negative subgroup was 78.35 ± 8.75 (Table 2).

p-S6 expression in more than 50% of tumor cells was observed in 61 cases (52.5%) (Table 3). p21 expression in at least 10% of tumor cells was noted in 96 cases (82.8%) (Table 3).

No correlation was found between p-S6 and p21 expression, based on a chi-square test (*p* = 0.230) and Cramer’s V test (*p* = 0.215) (Table 4).

p53 expression was positive in 90 patients (77.6%). Neither p-S6 nor p21 expression was significantly associated with p53 expression (*p* = 0.884 and *p* = 0.138, respectively).

As illustrated in Figure 1, p-S6 and p21 expressions had homogeneous patterns in the tumor cells.

When analyzing the prognostic significance of p-S6 expression in a univariate model, we found that p-S6 negative expression predicted a significantly greater risk of nodal metastasis (hazard ratio HR = 2.63, 95% confidence interval (CI), 1.51–4.54; *p* < 0.001) (Table 5). As previously described, in the univariate model, the factors showing a significant effect on metastasis risk are: tumor thickness, tumor diameter, desmoplasia, pathological stage, perineural infiltration, and the presence of tumor buds. These factors, along with p-S6 expression, were included in a multivariate model in which, finally, the loss of p-S6 expression proved to be a statistically significant independent predictor of the presence of metastasis (HR = 2.23, 95% CI, 1.01–4.91; *p* = 0.047). Tumor thickness and tumor buds were also significantly associated with metastasis and tumoral mortality. Age and tumor buds showed a significant impact on all-cause mortality (Table 5).

In contrast, p21 expression did not significantly predict an increased risk of nodal metastasis in the multivariate analysis (HR = 0.86, 95% CI, 0.45–1.66; *p* = 0.656). Thickness is the factor that showed the significant impact of metastasis and tumor mortality and age on all-causes mortality (Table 6).

### Survival Curves

As illustrated in Figure 2, significant differences in nodal metastasis-free survival (*p* < 0.001), tumoral mortality (*p* < 0.001) and overall survival (*p* = 0.006) rates were observed between the p-S6-positive and p-S6-negative patient subgroups.

As illustrated in Figure 3, a significant difference in the nodal metastasis-free rate (*p* = 0.018) was observed between the p21-positive and p21-negative patient subgroups.

As shown in Figure 2, there was a large difference in metastasis-free survival between the groups with high and low levels of p-S6 expression (*p* < 0.001). The patients with tumors with a low level of p-S6 expression also had a higher risk of tumor-specific and global mortality compared with tumors with a high level of p-S6 expression (*p* < 0.001 and *p* = 0.006, respectively). In the case of p21, differences in metastasis-free survival were found between the p21-positive and p21-negative groups (*p* = 0.018) that were not apparent for tumor-specific mortality (*p* = 0.269) or overall survival (*p* = 0.316) (Figure 3).

Statistically significant differences were also found when comparing p-S6 and p21 double-positive patients and their double-negative counterparts (Figure 4).

In the sample from the Marqués de Valdecilla University Hospital, the expression of less than 50% of p-S6 was associated with metastatic squamous cell carcinomas (Table 7).

## 3. Discussion

This study investigates the clinical significance of p-S6 and p21 expression as predictive factors for nodal metastases and survival in cSCCHN patients. We retrospectively analyzed p-S6 and p21 protein expression using a large homogenous cohort of cSCCHN patients. We chose these two proteins for analysis based on our group’s previous work [7,8,9].

It has been extensively demonstrated that S6 phosphorylation represents a critical downstream component of mTOR signaling, which has led to p-S6 levels frequently being used as a readout of mTOR activity. We provide original evidence demonstrating that negative p-S6 expression is an independent risk factor for nodal metastasis in cSCCHN patients. Published p-S6 studies have mainly taken a negative-versus-positive immunohistochemical approach (p-S6 ≤ 10% vs. >10%). In our study, p-S6 is divided into two groups of staining greater than 50% and less than or equal to 50%, because so few tumors exhibited values less than 10%. The expression of p-S6 in skin simples has not been thoroughly studied. It has been detected in Bowen’s disease and in cSCC with and without metastasis, but rarely in seborrheic keratosis, basal cell carcinoma or actinic keratosis [10,11]. p-S6 is found in 81% of cSCCs and in 64% of basocellular carcinomas. Neither study found the expression of p-S6 in normal skin, similar to our results. p-S6 expression is more widespread and intense in cSCC than in basocellular carcinomas. None of the studies have considered its role as a prognostic factor.

We only found one study of p-S6 and cSCCHN that considers their value as prognostic factors [12]. This study obtained different results from ours, whereby the expression of p-S6 was found to be associated with more frequent metastases in the parotid gland [12]. In this study, the number of patients studied was low (N = 37), a different antibody and antibody-measurement system were used, and the statistical methods employed differed from those used by us. For this reason, we requested a second population of metastatic and non-metastatic squamous cell carcinomas (22 patients) from the Pathological Anatomy Service of the Marqués de Valdecilla University Hospital. The results from this population corroborate our finding that the loss of expression is associated with metastatic carcinomas (chi-square test, *p* = 0.007).

Immunoexpression of p-S6 is associated with several cellular functions, including protein synthesis, mRNA processing, glucose homeostasis, cell growth, and survival [13].

The clinical impact of mTOR activation depends on the tumor type. In line with our results, recent meta-analyses have shown that the hyperactivation of mTOR is associated with a better prognosis, such as in non-small-cell lung cancer [14] and in luminal breast cancer [15]. However, it is associated with a worse prognosis in other tumor types, such as head and neck cancer [16], gastric carcinoma [17], low-grade glioma [18], renal cell carcinoma [19], and nasopharyngeal carcinoma [20].

The cytoplasmic expression of p-S6 has been shown to be a predictive factor for disease-free survival in hypopharyngeal [7] and laryngeal carcinomas [7,9] and in prostate adenocarcinoma [21], and it has been associated with smaller tumors in the oral cavity [8].

Regarding the expression of p21, in our study, we found that 82.8% of HNSCCs, 91.4% of NMSCs and 74.1% of MSCCs express p21. The expression of p21 in cutaneous squamous cell carcinoma has been very little studied [22].

We found no staining in normal skin controls, similar to what was described by Ahmed et al. [22], who found some cells in the stratum spinosum in non-photoexposed skin, but did not find them in photoexposed skin. Its presence has been noted in actinic keratoses, keratoacanthomas, cutaneous, and metastatic squamous cell carcinomas [22,23,24,25]. Lu et al. [26] found no differences in its expression between well- and poorly differentiated squamous cell carcinomas, similar to our results. Its overexpression has been described in patients with psoriasis or clogged skin, or whose skin has been exposed to irritants. The role of p21 in the induction and maintenance of cellular differentiation has been suggested [27]. Only in the univariate analysis was the loss of expression associated with an increase in the presence of lymph node metastases: it was not maintained in the multivariate model.

The results from our univariate analysis, in which the absence of p21 expression is associated with a greater number of metastases, are similar to findings in cancers of other locations such as the colon [28], pancreas [29], and breast [30]. However, this association was not maintained in the multivariate analysis.

Regarding the co-expression of the two proteins, we found no relationship between them using Cramer’s V test 0.115 (*p* = 0.215). This is a different outcome from those reported in pharyngeal [7] and oral [8] tumors.

We found statistically significant differences when studying the co-expression of the two proteins in the development of metastasis. This is similar to the findings of Llanos S et al. [7], who described the expressions of p21, p-S6, and the combination of the two, as being associated with greater disease-free survival in laryngeal and hypopharyngeal tumors, especially in patients with squamous cell carcinoma of the head and neck without lymphatic involvement at the time of diagnosis.

We did not find a relationship between the expressions of p53 and p-S6 and p21, probably because so many of the tumors expressed p53, as they arose in photo-exposed skin, in which the overexpression of this protein has been observed in elderly patients [31].

We are well aware that there are several limitations in this work. First, there are potential biases due to the retrospective nature of our study. Second, this study is limited to cSCCHN patients from a university hospital, and it therefore has a higher percentage of poor prognostic tumors than other hospitals because of referrals. Third, the lack of a standardized protocol for p-S6 protein evaluation and staining scoring limits the comparison of our findings with others. Fourth, our analysis is based on tissue microarrays, and hence protein scoring could not reflect the entire tumor. Nevertheless, we found highly concordant expression levels in the three representative tissue cores selected from each tumor. Fifth, the study was performed at a unique center. Sixth, a sample size calculation was not performed. Seventh, a second independent group of samples was used to validate the results obtained, but the sample size was small.

## 4. Materials and Methods

### 4.1. Patients and Procedures

The Department of Pathology’s electronic database at the Hospital Universitario Central de Asturias was searched to locate all the patients who had developed nodal metastases from cSCC of the head and neck (McSCCHN) between 1998 and 2017. Four of the authors (RS, BN, IF, and SR) read reports identified by this search. This is a case–control study on patients with head and neck cSCC treated with conventional surgery who have developed a histologically confirmed nodal metastasis. Control patients are defined as those with cSCCHN treated with conventional surgery who did not develop any metastasis and who had a minimum follow-up of five years. Cases and controls with affected margins in the resection specimen, and those who have received any adjuvant treatment after resection, were excluded. All the electronic medical records were reviewed to determine whether outcomes of interest had been achieved. Finally, 58 primary cSCCHN patients were included. Controls (58 patients) were randomly selected from those patients with cSCCHN who did not develop any metastases (cSCCHN). Ethics approval was obtained from the Hospital Universitario Central de Asturias Committee (2022-439). The study was conducted and the results reported in accordance with the Strengthening the Reporting of Observational Studies in Epidemiology guidelines for case–control studies. Clinical patient-related data were collected retrospectively. Patient age was taken as the age at the time of resection. Immunosuppressed patients included those with chronic lymphatic leukemia (3), liver transplant (1), kidney transplant (6), heart transplant (2), diffuse large lymphoma B cell (1), myeloma (1), inflammatory bowel disease treated with immunosuppressors (1) and polycythemia vera treated with hydroxyurea (1).

Pathological tumor staging based on the 8th AJCC classification was also recorded [32]. The outcome data are based on one tumor per patient.

Once the experiment was completed, upon analyzing our results, we discovered a discrepancy with the only published study on p-S6 as a prognostic factor [12]. Consequently, we requested a second sample from the Pathology Service of Marqués de Valdecilla University Hospital (Santander, GVMC). We replicated the experiment with 11 patients and 11 controls.

### 4.2. Histopathological Evaluation

The following histopathological features were analyzed and recorded for each sample using hematoxylin–eosin-stained slides: maximum diameter, tumor thickness (dichotomized as ≤6 mm or >6 mm; none less than 2 mm), anatomical level (Clark’s level), degree of histopathological differentiation (classified as well (1), moderately (2), or poorly (3) differentiated), presence of desmoplasia, perineural or perivascular invasion, and the presence and number of tumor budding events. Tumor budding is defined as the presence of either isolated single cells or small cell clusters (≤4) scattered in the stroma ahead of the invasive tumor front. The intensity of tumor budding (budding index) was classified as low (<5 buds) or high (≥5 buds) [33].

### 4.3. Tissue Microarray Construction

Morphologically representative areas were selected from each individual tumor paraffin block to construct a tissue microarray (TMA). Three 1 mm cylinders were taken to construct TMA blocks, as described previously [34]. Five TMAs were created, containing three tissue cores from each of the 116 cSCCHNs. In addition, each TMA included three cores of normal skin as an internal control.

### 4.4. Immunohistochemistry

The TMAs were cut into 3 μm sections and dried on Flex IHC microscope slides (DakoCytomation, Glostrup, Denmark). The sections were deparaffinized with standard xylene and hydrated through graded alcohols into water. Antigen retrieval was performed in all 121 samples using Envision Flex Target Retrieval solution, with a high pH (Dako), at 95 °C for 20 min. Endogenous peroxidase activity was suppressed by incubation for 5 min with 3% hydrogen peroxide (Dako). Staining was done at room temperature on an automatic staining workstation (Dako Autostainer, DakoCytomation) using the Dako Envision Flex 1 Visualization System (Dako Autostainer) and diaminobenzidine chromogen as substrate. The following primary antibodies were used: rabbit anti-phospho-S6 Ribosomal Protein (Ser235/236; Cell Signaling #2211) at 1:200 dilution; Novocastra Liquid mouse monoclonal antibody p21WAF1 protein (Clone 4D10; Leica Biosystems NCL-L-WAF-1, Barcelona, Spain) at 1:10 dilution, and mouse p53 protein (Clone DO-7; Dako #M 7001) at 1:10 dilution. The slides were viewed randomly, without clinical data, by two of the authors. The average intraobserver and interobserver variation was <5%. Immunostaining for p-S6 was evaluated on a semiquantitative scale (<10%, 10–50%, or >50% positive tumor cells). For statistical analysis, staining data were dichotomized as low expression (0–50% stained cells) or positive expression (>50% stained cells). Staining data for p21 and p53 were dichotomized as negative (0–10% stained cells) versus positive (>10% stained cells) expression.

### 4.5. Statistical Analysis

Baseline demographic and clinical characteristics of the patients and pathological data were summarized with standard descriptive statistics. The primary endpoints were time to lymph node metastasis and time to all-cause mortality, defined as the time from the date of diagnosis of the primary tumor to the date of diagnosis of metastasis, or of death for any cause, respectively. Tumor-specific deaths were also considered. Conventionally, depending on their symmetry and nature, variables are described as the mean ± standard deviation (SD), and the median with 25 and 75 percentiles.

The influence of factors on mortality was analyzed by standard proportional hazard Cox regression modeling. Raw and adjusted HRs and 95% confidence intervals were provided. Adjusted models include variables related to disease severity. Relapse was included as a time-dependent covariate. Probabilities of disease-free survival (nodal metastasis or death) and overall survival were estimated by the Kaplan–Meier approach. The correlation between variables has been studied and those that are highly related are excluded from the survival analysis.

All reported probabilities are 2-sided, and values of *p* < 0.05 were considered statistically significant. All analyses were carried out using IBM SPSS Statistics for Windows (Version 27.0; IBM Corp., Armonk, NY, USA).

## 5. Conclusions

This study reveals the independent prognostic relevance of p-S6 expression using a large homogeneous cohort of cSCCHN. The data presented reveal that p-S6 expression is a significant independent predictor for the risk of nodal metastasis in cSCCHN. p-S6 could serve as a biomarker to identify high-risk tumors at an early stage prior to metastasis. On the other hand, the expression of p21 was not found to be a significant independent predictor for nodal metastasis or survival outcomes in cSCCHN.

## Figures and Tables

**Figure 1 ijms-25-04304-f001:**
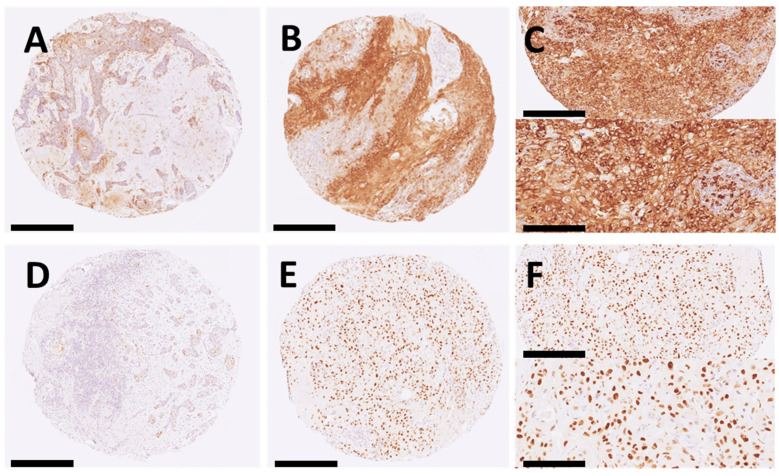
Immunohistochemical analysis of p-S6 and p21 expression in cSCC tissue specimens. (**A**) p-S6 ≤ 50% (×20 magnification; scale bar 250 µm); (**B**) p-S6 > 50% (×20 magnification; scale bar 250 µm); (**C**) p-S6 cytoplasmic stain (magnification of ×200 and ×400; scale bars of 100 µm and 50 µm); (**D**) p21 ≤ 10%; (**E**) p21 > 10% (×20 magnification; scale bar 250 µm); (**F**) p21 nuclear stain (magnification of ×200 and ×400); scale bars of 100 µm and 50 µm).

**Figure 2 ijms-25-04304-f002:**
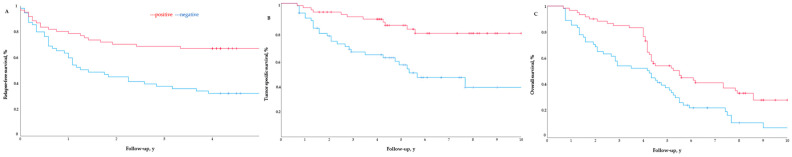
Kaplan–Meier survival estimates of nodal metastasis-free survival (**A**), tumor-related death (**B**) and overall survival (**C**) after resection of cutaneous squamous cell carcinoma for the p-S6 groups. p-S6 positivity was defined as >50% tumor cell staining positive for p-S6.

**Figure 3 ijms-25-04304-f003:**
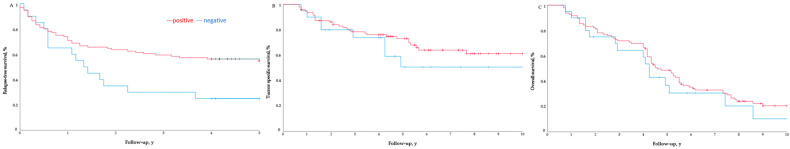
Kaplan–Meier survival estimates of nodal metastasis-free survival (**A**), tumor-related death (*p* = 0.269) (**B**) and overall survival (*p* = 0.316) (**C**) after resection of cutaneous squamous cell carcinoma for the p21 groups. p21 positivity was defined as >10% tumor cells staining positive for p21.

**Figure 4 ijms-25-04304-f004:**
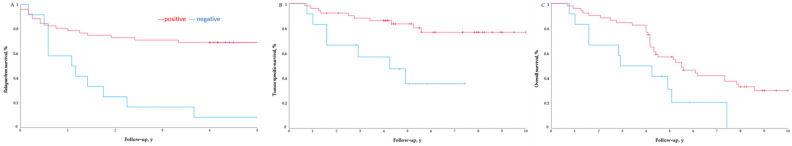
Kaplan–Meier survival estimates of nodal metastasis-free survival (*p* < 0.001) (**A**), tumor-related death (*p* = 0.002) (**B**) and overall survival (*p* = 0.024) (**C**) after resection of cutaneous squamous cell carcinoma for the p21 and p-S6-positive and p21 and p-S6-negative groups.

**Table 1 ijms-25-04304-t001:** Association between p-S6 expression and clinicopathological features of patients with primary cutaneous squamous cell carcinoma.

		p-S6 Expression		
Clinicopathological Characteristics	N = 116	Negative (≤50%) N = 55; n (%)	Positive (>50%) N = 61; n (%)	*p*
Sex, male	89	42 (76.3)	47 (77.0)	1.000
Age, mean ± SD, (min–max)	78.74 ± 8.32 (50–97)	79.11 ± 7.73 (51–93)	78.41 ± 8.87 (50–97)	0.713
Tumor thickness, mm, mean ± SD	8.91 ± 6.54	11.55 ± 7.38	6.54 ± 4.59	**<0.001**
>6 mm	60	40 (72.7)	20 (32.8)	**<0.001**
Tumor horizontal size, mm, mean ± SD	21.56 ± 13.89	25.27 ± 16.09	18.21 ± 10.63	**0.007**
>20 mm	42	29 (52.7)	13 (21.3)	**<0.001**
Tumor differentiation				0.110
Well differentiated	61 (52.6%)	27 (49.1)	34 (55.7)	
Moderately differentiated	48 (41.4%)	22 (40.0)	26 (42.6)	
Poorly differentiated	7 (6.0%)	6 (10.9)	1 (1.6)	
Desmoplastic growth	21	15 (27.3)	6 (9.8)	**0.017**
Tumor site, ear	26	12 (21.8)	14 (22.9)	1.000
pTNM 8th edition				**0.007**
pT1	64 (55.2%)	22 (40.0)	42 (68.9)	
pT2	36 (31.0%)	22 (40.0)	14 (23.0)	
pT3	16 (13.8%)	11 (20.0)	5 (8.2)	
Inflammation				0.609
None	29	15 (27.3)	14 (23.0)	
Mild	73	35 (63.6)	38 (62.2)	
Strong	14	5 (9.1)	9 (14.8)	
Other SCC	38	13 (23.6)	25 (40.1)	0.051
Immunosuppression				1.000
Yes	16 (13.8%)	8 (14.5)	8 (13.1)	
No	100 (86.2%)	47 (85.5)	53 (86.9)	
Perineural invasion				**0.029**
Yes	16 (13.8%)	12 (21.8)	4 (6.6)	
No	100 (86.2%)	43 (78.2)	57 (93.4)	
Lymph-vascular invasion				0.603
Yes	3 (1.6%)	2 (3.6)	1 (1.6)	
No	113 (97.4%)	53 (96.4)	60 (98.4)	
Tumor buds present				**<0.001**
Yes	54	37 (67.3)	17 (27.9)	
No	62	18 (32.7)	44 (72.1)	
≥5 tumor buds				0.258
Yes	24	14 (25.5)	10 (16.4)	
No	92	41 (74.5)	51 (83.6)	

**Table 2 ijms-25-04304-t002:** Association between p21 expression and clinicopathological features of patients with primary cutaneous squamous cell carcinoma.

		**p21 Expression**		
**Clinicopathological Characteristics**	**N = 116**	**Negative N = 20** **n (%)**	**Positive N = 96** **n (%)**	** *p* **
Sex, male	89	16 (18.0)	73 (82.0)	1.000
Age, mean ± SD (min–max)	78.74 ± 8.32 (50–97)	78.35 ± 8.75 (54–96)	78.82 ± 8.27 (50–97)	0.818
Tumor thickness, mm, mean ± SD	8.91 ± 6.54	11.30 ± 6.02	8.42 ± 6.57	0.869
>6 mm	60	15 (75.0)	45 (46.9)	**0.027**
Tumor horizontal size, mm, mean ± SD	21.56 ± 13.89	21.05 ± 9.66	21.67 ± 14.66	0.113
>20 mm	42	8 (40.0)	34 (35.4)	0.799
Tumor differentiation				0.110
Well differentiated	61	27 (49.1)	34 (55.7)	
Moderately differentiated	48	22 (40.0)	26 (42.6)	
Poorly differentiated	7	6 (10.9)	1 (1.6)	
Desmoplastic growth	21	15 (27.3)	6 (9.8)	**0.017**
Tumor site, ear	26	3 (15.0)	23 (23.9)	0.558
pTNM 8th edition				0.329
pT1	64	9 (45.0)	55 (57.3)	
pT2	36	9 (45.0)	27 (28.1)	
pT3	16	2 (10.0)	14 (14.6)	
Inflammation				0.368
None	29	7 (35.0)	22 (22.9)	
Mild	73	12 (60.0)	61 (63.5)	
Strong	14	1 (5.0)	13 (13.5)	
Other SCC	38	5 (25.0)	33 (34.4)	0.601
Immunosuppression				0.474
Yes	16	4 (20.0)	12 (12.5)	
No	100	16 (80.0)	84 (87.5)	
Perineural invasion				**0.032**
Yes	16	6 (30.0)	10 (10.4)	
No	100	14 (70.0)	86 (89.6)	
Lymph-vascular invasion				0.436
Yes	3	1 (5.0)	2 (2.08)	
No	113	19 (95.0)	94 (97.9)	
Tumor buds present				0.222
Yes	54	12 (60.0)	42 (43.8)	
No	62	8 (40.0)	54 (56.2)	
≥5 tumor buds				0.361
Yes	24	6 (3.0)	18 (18.8)	
No	92	14 (70)	78 (81.2)	

**Table 3 ijms-25-04304-t003:** p-S6 and p21 expression in cSCCHN and metastatic cSCCHN (McSCCHN).

p-S6	<10%	10–50%	>50%	*p* (Chi-Square)
**cSCCHN**	0 (0%)	17 (29.3%)	41 (70.7%)	**<0.001**
**McSCCHN**	4 (6.9%)	34 (58.6%)	20 (34.5%)	
**p21**	**0–10%**		**>10%**	***p* (chi-square)**
**cSCCHN**	5 (8.6%)		53 (91.4%)	**0.025**
**McSCCHN**	15 (25.9%)		43 (74.1%)	

**Table 4 ijms-25-04304-t004:** Correlation between p-S6 and p21 expression.

	p-S6 ≤ 50	p-S6 > 50	*p* (Chi-Square)	*p* (Cramer’s V)
**p21 (−)**	12 (60.0%)	8 (40.0%)	0.230	0.215
**p21 (+)**	43 (44.8%)	53 (55.2%)		

**Table 5 ijms-25-04304-t005:** Univariate and multivariate models for the effects of p-S6 expression on nodal metastasis, tumor mortality and all-cause mortality.

	Metastasis		Tumor Mortality		All-Cause Mortality	
p-S6	HR (95% CI)	*p*	HR (95% CI)	*p*	HR (95% CI)	*p*
			**Univariate**			
p-S6	2.63 (1.51–4.54)	**<0.001**	3.70 (1.78–7.69)	**<0.001**	1.75 (1.14–2.63)	**0.008**
			**Multivariate**			
p-S6	2.23 (1.01–4.91)	**0.047**	0.99 (0.42–2.29)	0.975	1.08 (0.66–1.77)	0.766
Age	0.99 (0.65–2.31)	0.526	1.01 (0.97–1.07)	0.975	1.05 (1.01–1.09)	**0.005**
Sex	0.79 (0.55–2.18)	0.791	1.20 (0.49–2.91)	0.684	0.88 (0.54–1.47)	0.634
Tumor thickness	3.25 (1.55–6.85)	**0.002**	4.03 (1.42–11.42)	**0.009**	1.14 (0.66–1.99)	0.642
Tumor horizontal size	1.12 (0.48–2.58)	0.798	0.83 (0.27–2.52)	0.743	0.97 (0.43–2.20)	0.934
Desmoplastic growth	1.08 (0.55–2.13)	0.812	0.88 (0.36–2.11)	0.771	0.85 (0.45–1.63)	0.632
Perineural invasion	1.27 (0.58–2.79)	0.547	0.80 (0.28–2.29)	0.802	1.12 (0.53–2.32)	0.770
Tumor buds	6.72 (3.32–13.58)	**<0.001**	7.93 (3.12–20.17)	**<0.001**	3.36 (2.09–5.42)	**<0.001**
pTNM 8th edition	1.44 (0.87–2.36)	0.152	2.13 (1.20–4.06)	**0.021**	1.95 (1.18–3.24)	**0.009**

**Table 6 ijms-25-04304-t006:** Univariate and multivariate models for the effect of p21 expression on nodal metastasis, tumor mortality, and all-cause mortality.

	Metastasis		Tumor Mortality		All-Causes Mortality	
p21	HR (95% CI)	*p*	HR (95% CI)	*p*	HR (95% CI)	*p*
			**Univariate**			
p21	4.76 (1.11–5.55)	**0.020**	1.72 (0.77–3.84)	0.185	1.31 (0.71–2.27)	0.323
			**Multivariate**			
p21	0.86 (0.45–1.66)	0.656	1.14 (0.48–2.73)	0.770	0.72 (0.39–1.33)	0.720
Age	0.99 (0.97–1.03)	0.945	1.03 (0.98–1.08)	0.207	1.03 (1.01–1.07)	**0.014**
Sex	1.13 (0.57–2.22)	0.727	1.37 (0.57–3.33)	0.482	0.96 (0.58–1.60)	0.885
Tumor thickness	4.31 (2.23–8.32)	**<0.001**	7.00 (2.80–17.53)	**<0.001**	1.79 (1.12–2.84)	**0.014**
Desmoplastic growth	1.48 (0.76–2.88)	0.249	1.62 (0.69–3.79)	0.266	1.00 (0.54–1.87)	0.999
Perineural invasion	1.52 (0.76–3.05)	0.234	1.16 (0.48–2.80)	0.739	1.45 (0.73–2.87)	0.285

**Table 7 ijms-25-04304-t007:** Expression of p-S6 in cSCCHN and McSCCHN patients from the Marqués de Valdecilla University Hospital sample.

p-S6	≤10%	10–50%	>50%	*p* (Chi-Squared)
cSCCHN	0 (0%)	1 (29.3%)	10 (70.7%)	0.007
McSCCHN	4 (6.9%)	3 (58.6%)	4 (34.5%)	

## Data Availability

The data presented in this study are available on request from the corresponding author (accurately indicate status).

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
