# Peer review of "Low-Level Expression of p-S6 Is Associated with Nodal Metastasis in Patients with Head and Neck Cutaneous Squamous Cell Carcinoma"

_ijms, 2024, doi:10.3390/ijms25084304_

Round 1

Reviewer 1 Report

Comments and Suggestions for Authors

The manuscript “Low-level expression of P-S6 is associated with nodal metastasis in patients with head and neck cutaneous squamous cell carcinoma” is a research article about P-S6 and p21 immunohistochemical analyses in cSCCHN patients and associations with clinical characteristics.

The manuscript has some flaws that must be addressed.

1.       Language revision is needed.

2.       The manuscript seems to be prepared carelessly. There are some gray parts, sometimes the text is double-spaced sometimes not. In a table, SD is written in bold without reason. Typos are present.

3.       In tables it should be reported also the median age, min and max.

4.       Please report in a box plot graph the findings about P-S6 and p21 expression.

5.       Please add bar size to figure 1.

6.       A crucial point is that authors should discuss more about the differential pathophysiology of actinic keratosis and squamous cell carcinoma, since squamous cell carcinoma often arises from actinic keratosis (PMID: 34302630). How about P-S6 and p21 expression in actinic keratosis? If immunohistochemical analyses cannot be performed, authors should at least discuss this crucial point.

Comments on the Quality of English Language

Extensive editing of English language required

Author Response

Thank you very much for your comments, which have helped us significantly improve the work.

The manuscript “Low-level expression of P-S6 is associated with nodal metastasis in patients with head and neck cutaneous squamous cell carcinoma” is a research article about P-S6 and p21 immunohistochemical analyses in cSCCHN patients and associations with clinical characteristics.

The manuscript has some flaws that must be addressed.

  1. Language revision is needed. Language revision has been completed by a native professional reviewer
  2. The manuscript seems to be prepared carelessly. There are some gray parts, sometimes the text is double-spaced sometimes not. In a table, SD is written in bold without reason. Typos are present.

Thank you for bringing these issues to our attention. We have carefully reviewed the text to address inconsistencies and typos

  1. In tables it should be reported also the median age, min and max.

We include this information in the revised version

  1. Please report in a box plot graph the findings about P-S6 and p21 expression.

Unfortunately, we are unable to create a box plot graph because the evaluation of the patient´s slides was conducted by categorizing them as qualitative variables rather than quantitative. As stated in the methods section “The average intraobserver and interobserver variation was <5%. Immunostaining for p-S6 was evaluated on a semiquantitative scale (<10%, 10-50%, or >50% positive tumor cells)”.

This method was consistently chosen in all our articles due to its higher inter-observer correlation and greater ease of reproducibility

If the reviewer prefers, we can include the following graph as supplementary material. However, these data are collected in Table 4.

  1. Please add bar size to figure 1. Bar size has been added to figure 1, and the corresponding text has been adjusted accordingly to provide clarity

Figure 1.  Immunohistochemical analysis of P-S6 and p21 expression in cSCC tissue specimens. A, P-S6 ≤50% (x20 magnification; scale bar 250µm); B, P-S6>50% (x20 magnification; scale bar 250µm); C, P-S6 cytoplasmic stain (magnification of x200 and x400; scale bars of 250 µm and 50 µm); D, p21≤10% ; E, p21>10% (x20 magnification; scale bar 250µm); F, p21 nuclear stain (magnification of x200 and x400); scale bars of 100 µm and 50 µm).

  1. A crucial point is that authors should discuss more about the differential pathophysiology of actinic keratosis and squamous cell carcinoma, since squamous cell carcinoma often arises from actinic keratosis (PMID: 34302630). How about P-S6 and p21 expression in actinic keratosis? If immunohistochemical analyses cannot be performed, authors should at least discuss this crucial point.

We appreciate your comment as it highlights the potential for studying the expresión of these moleecules in actinic keratosis. However, it´s important to note that our investigation did not include the study of actinic Keratosis.

Furthermore, it is worth mentioning that there is limited literature available on this topis, as we have discussed in the discussion section:

“The expression of p-S6 in skin simples has not been thoroughly studied. It has been detected in Bowen’s disease and in cSCC with and without metastasis, but rarely in seborrheic keratosis, basal cell carcinoma or actinic keratosis [10,11]. P-S6 is found in 81% of cSCCs and in 64% of basocellular carcinomas. Neither study found expression of p-S6 in normal skin, similar to our results. P-S6 expression is more widespread and intense in cSCC than in basocellular carcinomas. None of the studies has considered its role as a prognostic factor.”

The article titled is indeed interesting, but it lacks data overlap with our study. Additionally, there is abundant evidence demonstrating changes in the expresión of proteins, RNA, and DNA in both actinic keratosis and squamous cell carcinomas, which we have chosen not to cite in this manuscript given that it is not the focus of this investigation

Reviewer 2 Report

Comments and Suggestions for Authors

The work of the authors on the prognostic role of p21 and pS6 in skin SCC.

I have only some minor points to raise:

- how were the cut-offs for IHC evaluation chosen? I mean, why did you choose in a case a cut-off of 10% and in the other 50%? Are there other previous studies indicating such values? It is a point that you could expand and clarify in the methods

- please check the manuscript throughout for typos and words in Spanish/English in the tables

- the tables could be improved in readability by highlighting the significant parameters, and in the tables on survival analysis it is not clear if the several parameters are reported both for univariate and for multivariate analysis

- in the methods, please add how you dealt with collinearity in survival analysis

Comments on the Quality of English Language

Only minor polishing of English is advised.

Author Response

Thank you very much for your comments, which have helped us significantly improve the work.

I have only some minor points to raise:

- how were the cut-offs for IHC evaluation chosen? I mean, why did you choose in a case a cut-off of 10% and in the other 50%? Are there other previous studies indicating such values? It is a point that you could expand and clarify in the methods

Thanks you for your comment:

The cutoff points selected for Ps6 are differ from those commonly used in the literature. We opted for a cutoff of 50% instead of the typical 10% to ensure comparability between the two groups. For p21, we adhered to the standard cutoff commonly used in this type of research.

We have stated in the text methods:

The average intraobserver and interobserver variation was <5%. Immunostaining for p-S6 was evaluated on a semiquantitative scale (<10%, 10-50%, or >50% positive tumor cells). For statistical analysis, staining data were dichotomized as low expression (0-50% stained cells) or positive expression (>50% stained cells). Staining data for p21 and p53 were dichotomized as negative (0-10% stained cells) versus positive (>10% stained cells) expression.

And in the discusión:

We provide original evidence demonstrating that p-S6 negative expression is an independent risk factor for nodal metastasis in cSCCHN patients. Published p-S6 studies have mainly taken a negative-versus-positive immunohistochemical approach (p-S6 ≤10% vs. >10%). In our study, p-S6 is divided into two groups with staining greater than 50% or less than or equal to 50% because so few tumors exhibited values less than 10%.

- please check the manuscript throughout for typos and words in Spanish/English in the tables

We have done it

- the tables could be improved in readability by highlighting the significant parameters, and in the tables on survival analysis it is not clear if the several parameters are reported both for univariate and for multivariate analysis

Thank you for your feedback. We have worked on improving the readability of the tables by highlighting significant parameters. Additionally, we ensured clarity regarding the reporting of several parameters for both univariate and multivariate analysis in the survival analysis tables. We appreciate your suggestions for enhancing the presentation of our data.

- in the methods, please add how you dealt with collinearity in survival analysis

Thank  you for your comment, We have addressed it by adding the following statement in the Methods section

The correlation between variables has been studied, and those that are highly related have been excluded from the survival analysis

Reviewer 3 Report

Comments and Suggestions for Authors

I find the manuscript hard to follow because of the order of the sections. This is probably the way the journal wants it so that is no criticism to the authors.

However, there are some things missing or hard to understand in the methods section that makes it hard to assess the results presented.

Patient selection

1.       How were the 58 patients with nodal metastases included? This is not described clearly in the methods section and is important to know when the results are assessed. A flowchart would be helpful.

2.       There were only approximately 3 pats/year included in this group (or were they selected). Were there other patients excluded?

3.       Why were patients with positive margins excluded? If p-S6 is a risk factor for nodal metastasis it could help to prioritize re-excision.

4.       The inclusion and exclusion criteria should be clearly stated.

5.       Was any sample size analysis performed?

6.        What is meant: “Given the discrepancy with the only published study of p-S6 as a prognostic factor [12], a 307 second population was sought, made up of 11 patients and controls from the Pathology Service of 308 the Marqués de Valdecilla University Hospital (Santander, GVMC).”

Immunohistochemistry

1.       The groups defined above or below a threshold for p-S6 (50 %) and p21 (10 %). Are these thresholds chosen arbitrarily or generally accepted?

Results

1.       The red and blue groups should be defined IN the Kaplan-Meier curves not just in the text.

Comments on the Quality of English Language

Could use some editing

Author Response

Thank you very much for your comments, which have helped us significantly improve the work.

Patient selection

  1. How were the 58 patients with nodal metastases included? This is not described clearly in the methods section and is important to know when the results are assessed. A flowchart would be helpful.

We have added in the text: Four of the authors (RS, BN, IF, and SR) reviewed reports identified by this search.

We likely don't grasp the reviewer's intended meaning.

Given the current data available, it would be unfeasible to produce a flowchart within such a limited timeframe. It's a work of years. At this moment, we only have data from the cases and controls

As stated in the first article of our series, the methodology description is identical to that found in:

1: Santos-Juanes J, Fernández-Vega I, Lorenzo-Herrero S, Sordo-Bahamonde C, Martínez-Camblor P, García-Pedrero JM, Vivanco B, Galache-Osuna C, Vazquez-Lopez F, Gonzalez S, Rodrigo JP. Lectin-like transcript 1 (LLT1) expression is associated with nodal metastasis in patients with head and neck cutaneous squamous cell carcinoma. Arch Dermatol Res. 2019 Jul;311(5):369-376. doi: 10.1007/s00403-019-01916-x. Epub 2019 Apr 6. PMID: 30955082.

2: Munguía-Calzada P, Fernández-Vega I, Martínez-Camblor P, Díaz-Coto S, García-Pedrero JM, Vivanco B, Osuna CG, Vazquez-Lopez F, Rodrigo JP, Santos-Juanes J. Correlation of focal adhesion kinase expression with nodal metastasis in patients with head and neck cutaneous squamous cell carcinoma. Head Neck. 2019 May;41(5):1290-1296. doi: 10.1002/hed.25556. Epub 2018 Dec 10. PMID: 30537291.

3: García-Pedrero JM, Martínez-Camblor P, Diaz-Coto S, Munguia-Calzada P, Vallina-Alvarez A, Vazquez-Lopez F, Rodrigo JP, Santos-Juanes J. Tumor programmed cell death ligand 1 expression correlates with nodal metastasis in patients with cutaneous squamous cell carcinoma of the head and neck. J Am Acad Dermatol. 2017 Sep;77(3):527-533. doi: 10.1016/j.jaad.2017.05.047. Epub 2017 Jul14. PMID: 28716437.

4: Gonzalez-Guerrero M, Martínez-Camblor P, Vivanco B, Fernández-Vega I, Munguía-Calzada P, Gonzalez-Gutierrez MP, Rodrigo JP, Galache C, Santos-Juanes J. The adverse prognostic effect of tumor budding on the evolution of cutaneous head and neck squamous cell carcinoma. J Am Acad Dermatol. 2017 Jun;76(6):1139-1145. doi: 10.1016/j.jaad.2017.01.015. Epub 2017 Mar 14. PMID:28314684.

  1. 2.There were only approximately 3 pats/year included in this group (or were they selected). Were there other patients excluded?

As we have pointed out in the text, patients who received adjuvant treatments or who had affected edges were excluded, as were those who were lost to follow-up.

  1. Why were patients with positive margins excluded? If p-S6 is a risk factor for nodal metastasis it could help to prioritize re-excision.

      Your comment is very interesting. However, in line with the scope of our work outlined above, we have decided to exclude patients with affected edges. This decision is based on literature suggesting that such cases could introduce a potential confounding factor. Additionally, many of these patients were excluded due to receiving supplementary radiotherapy treatment.

  1. The inclusion and exclusion criteria should be clearly stated.

We think that is is clear in the text, but it the referee prefers we can write

      CASE: inclusión criteria: patients who developed histologically confirmed lymph node metastasis from cSCCHN.

      Exclusion criteria: patients who developed histologically confirmed lymph node metastasis from cSCCHN, with positive margins, and/or adjuvant therapy after the surgery

      Control: patients with cSCCHN who did not develop any metastases (cSCCHN) and who had a minimum follow-up of 5 years.

  1. Was any sample size analysis performed? No, it is a retrospective study.
  2. What is meant: “Given the discrepancy with the only published study of p-S6 as a prognostic factor [12], a 307 second population was sought, made up of 11 patients and controls from the Pathology Service of 308 the Marqués de Valdecilla University Hospital (Santander, GVMC).”

 Thank you for your comment, we didn't know where to indicate the following:

Once the experiment was finished, when we analyzed our results, we found a discrepancy with the only published study of p-S6 as a prognostic factor [ Ref 12]. For this reason we request a second sample from the Pathology Service of the Marqués de Valdecilla University Hospital (Santander, GVMC). We repeat the experiment in 11 patients and 11 controls.

Immunohistochemistry

  1. The groups defined above or below a threshold for p-S6 (50 %) and p21 (10 %). Are these thresholds chosen arbitrarily or generally accepted?

In most studies, a percentage higher than 10% is accepted; however, in the case of p-S6, the number of cases was very low, so we selected 50%, believing it to be successful as there are results that differentiate the prognosis.

In the original text we describe: “Published p-S6 studies have mainly taken a negative-versus-positive immunohistochemical approach (p-S6 ≤10% vs. >10%). In our study, p-S6 is divided into two groups with staining greater than 50% or less than or equal to 50% because so few tumors exhibited values less than 10%.”

Results

  1. The red and blue groups should be defined IN the Kaplan-Meier curves not just in the text.

Thank you very much for the correction; we have already updated the figure accordingly.

Round 2

Reviewer 1 Report

Comments and Suggestions for Authors

Unfortunately, the authors failed to answer to most of the concerns raised by the reviewer. I must recommend the rejection of the manuscript.

Comments on the Quality of English Language

Extensive editing of English language required

Author Response

All possible changes have been made; some of the requested ones are not within the scope of our work. Thank you for your feedback.

Reviewer 3 Report

Comments and Suggestions for Authors

Thank you for your answers that helped clarify some questions in the first draft. 

Author Response

Thank you for your feedback